# Assisted Reproduction Therapy in Patients with Multiple Sclerosis: Narrative Review and Practical Recommendations

**DOI:** 10.3390/healthcare13233155

**Published:** 2025-12-03

**Authors:** Lenka Mekiňová, Iva Šrotová, Petra Hanáková, Pavlína Danhofer, Robert Hudeček, Michal Ješeta

**Affiliations:** 1Department of Gynecology and Obstetrics, University Hospital Brno, 625 00 Brno, Czech Republic; 2Faculty of Medicine, Masaryk University Brno, 625 00 Brno, Czech Republic; 3Department of Neurology, University Hospital Brno, 625 00 Brno, Czech Republic; 4Department of Neurology, St. Anne’s University Hospital in Brno, 602 00 Brno, Czech Republic; 5Department of Pediatric Neurology, University Hospital Brno, 625 00 Brno, Czech Republic

**Keywords:** IVF, multiple sclerosis, autoimmune disease, ovarian stimulation, antagonist protocol, agonist protocol

## Abstract

**Objective**: The objective of this study is to present contemporary findings regarding the relationship between the application of assisted reproduction methods and their impact on the incidence of multiple sclerosis. **Design**: This study adopts a narrative review design. **Text**: Assisted reproductive technology (ART) is increasingly used to treat human infertility. Due to the massive use of these techniques, it is increasingly important to record not only the course of fertilization and embryonic and fetal development of the individual but also the overall health status of the children born and their mothers. The incidence of autoimmune diseases continues to rise for reasons that remain unclear. One of the factors considered in connection with autoimmune disorders is ART. Opinions on the safety and reliability of ART methods are not consistent. Recently, extensive studies focusing on this issue have been presented and have not found a connection between infertility treatment with assisted reproductive techniques and the development of multiple sclerosis (MS). **Conclusions**: Current evidence suggests that, in adherence to the principles of evidence-based medicine and modern approaches to multiple sclerosis therapy, assisted reproduction in women with this disease is effective and does not pose a serious health risk. Therefore, it is necessary to always individualize therapy with regard to future pregnancy. Interdisciplinary cooperation on the timing of IVF therapy and minimizing the risk of MS exacerbation is also important.

## 1. Introduction

Infertility affects approximately 17.5% of adult couples worldwide [1]. Global demographic data indicate a steadily declining fertility rate, which, together with the increasing age of women at first conception and significant advances in assisted reproductive technologies (ART), is increasing the demand for infertility services. The postponement of motherhood to a later age and the increased availability of assisted reproduction methods are causing a global rise in their use [2]. The use of assisted reproductive techniques is increasing worldwide. In Europe, 2 to 6% of children (in Czech republic 4%) are born each year via in vitro fertilization (IVF) [3]. There are many negative influences that impair fertility, but maternal age is one of the most significant. It has been repeatedly proven that with increasing maternal age, not only does the number of eggs decrease, but their quality also significantly decreases. This leads not only to worse fertilization of oocytes, but also to impaired development of early embryos, leading to early miscarriages [4].

Multiple sclerosis (MS) is a chronic autoimmune disease with a rising incidence that most commonly develops between the ages of 20 and 40 years, although it can also affect children and older patients. As with all autoimmune diseases, MS occurs significantly more frequently in women than in men, at a ratio of approximately 3:1 [5]. In women, the disease often manifests between the ages of 20 and 30, a period during which they address issues related to the menstrual cycle, choose appropriate contraception, plan for pregnancy, and navigate the course of pregnancy, postpartum periods, and breastfeeding. Hormonal fluctuations (including minor hormonal changes during the menstrual cycle, as well as those during pregnancy and the postpartum period) have been repeatedly associated with changes in MS activity, highlighting the influence of the endocrine system on immune processes [6].

Women with MS who are planning a pregnancy must wait for a period of disease stabilization and low radiological activity [7]. Consequently, these patients often have to postpone their attempts at pregnancy. Therefore, patients with MS are often older than their healthy counterparts when planning pregnancy [8]. Due to this fact, women with MS are more likely to require ART than healthy individuals [9]. The actual risk-benefit ratio of pregnancy should always be assessed on an individual basis. The monitoring requirements of patients with MS, coupled with the inherent complexity of pregnancy, lead to interdisciplinary collaboration, where a comprehensive and systematic approach to the care of patients with MS in their quest for motherhood is essential. Unfortunately, there is a lack of recommendations for the management of infertility in these patients from a multidisciplinary gynecological and neurological perspective.

This Narrative Review aimed to present recommendations from a group of neurologists, gynecologists, and embryologists for managing individuals with MS who wish to become parents. In this regard, we thoroughly examined the process, covering everything from preconception counseling to in vitro fertilization. One of the reasons for this review is the lack of specific guidelines and generally valid recommendations for ART in patients with multiple sclerosis. Additionally, we propose strategies for establishing and enhancing multidisciplinary units to care for patients with these conditions.

## 2. Methods

A multidisciplinary working group of six experts, comprising three neurologists specializing in multiple sclerosis, one embryologist, and two gynecologists specializing in assisted reproductive technology, convened multiple times to formulate recommendations concerning ART in patients with MS and to collaborate in the literature review. Prior to the consensus meetings, a comprehensive review of the extant literature was conducted using the Web of Science and PubMed databases, between years 1999–2025, by using the following keywords: multiple sclerosis OR demyelination disease AND fertility OR in vitro fertilization OR assisted reproductive technology OR gonadotropins OR ovarian stimulation OR reproduction. Only English-written articles were considered, and non-peer-reviewed studies were excluded. Duplicates were removed manually. The review included all observational studies, case reports, and expert opinions that assessed reproductive issues in patients with multiple sclerosis, with a particular focus on assisted reproductive techniques in this group of patients. Two investigators (L.M. and M.J.) independently screened the titles, keywords and abstracts for relevant indicators. The other authors subsequently reviewed the selected abstracts and excluded irrelevant studies. The manuscripts that were included, along with all identified research, underwent a second round of full-text screening to determine which entries qualified for inclusion. At the end of the selection process, 122 articles were included as references for the final recommendations. A flow diagram of the study is presented in Figure 1.

During the initial meetings, critical questions regarding infertility and ART in MS patients were identified and deliberated. The primary topics included the etiopathogenesis of MS, impact of MS on reproduction, risks associated with ART in MS, influence of sex hormones on the development and course of MS, optimal period for MS stabilization prior to undergoing ART and attempting pregnancy, management of disease-modifying therapy (DMT) in patients undergoing ART, recommended procedure for ovarian stimulation in women with MS, and recommended age for ART application. The collective consensus is encapsulated in the conclusion of this study. For each key topic, a recommendation was adopted after reaching a consensus. The information derived from the literature review and expert opinions gathered during joint sessions formed the foundation for this article, in which all authors participated. This article is based on an interdisciplinary discussion and presentation of previously conducted studies and does not include any results from new studies involving human participants or animals conducted by the authors.

This is a consensual Narrative Review based on data obtained from the published literature and the personal experiences of all authors. This review was carried out in accordance the Declaration of Helsinki [10] and respects the European GDPR [11].

### 2.1. Etiopathogenesis of Multiple Sclerosis

Multiple sclerosis is a chronic disease of the central nervous system characterized by damage to the myelin sheaths and axons in the central nervous system [12]. The pathogenesis of this disease is multifactorial and includes both genetic predisposition and environmental influences [13,14].

In Europe, the annual incidence varies by region from 6 to 19.5 per 100,000 per year [15]. In the children of patients with MS, the risk of developing MS is approximately 3%. Heredity is polygenic, and there is a notable association between polymorphisms in the HLA-DRB1*15:01 region [13]. Sex linkage is also significant, with women being affected more frequently.

External factors influencing the development of MS include vitamin D deficiency, which, in addition to its effect on the skeletal system, has a significant impact on the immune system, as evidenced by a higher incidence of MS in countries with low levels of UV radiation [16]. Another risk factor is contact with the Epstein–Barr virus (EBV), especially late primary infection, and high antibody titers against EBNA-1 [17]. Obesity during childhood and adolescence (particularly in girls), a diet high in salt (a pro-inflammatory effect leading to Th17 activation), changes in the gut microbiome (disrupting tolerance and promoting a pro-inflammatory environment), and frequent viral infections in early childhood (which may affect the development of immune tolerance and the proper balance of immune responses) also pose risks. Exposure to disease triggers and the interaction between genotype and environment play crucial roles [18].

The immunopathogenesis of multiple sclerosis is driven mainly by several fundamental processes, which essentially blend into a continuum, and at certain stages of the disease, one may slightly predominate over the others.

Peripheral Activation of the Immune System

The disease begins peripherally with the activation of autoreactive T-lymphocytes. It is imperative not to underestimate the crucial function of B lymphocytes, for which highly effective therapeutic interventions are currently available [19]. One possible pathological reaction likely develops due to a mechanism known as molecular mimicry, in which, for example, after an EBV infection, the immune system may mistakenly identify self-structures in the CNS that resemble viral antigens (e.g., myelin proteins—MBP and MOG) [20]. Several cytokines (such as IL-17, IFN-γ, TNF-α) also play a role in the pathophysiology, enabling the disruption of the blood–brain barrier (BBB) and the passage of pro-inflammatory agents into the CNS [21].

2.Inflammation and Damage in the Central Nervous system; the multiple sclerosis continuum: From Inflammation to Neurodegeneration

After crossing the BBB, activated T-lymphocytes (CD8^+^ and CD4^+^) and B-lymphocytes migrate into the CNS, where they participate in both inflammation and neurodegeneration, together with microglia [22]. Previously, MS was considered a disease with clearly separated phases of inflammation and neurodegeneration, which was also reflected in the preferred division into relapsing-remitting (RRMS), secondary progressive (SPMS), and primary progressive (PPMS) phases [23]. Today, a model of a pathological continuum is preferred, in which the process of neurodegeneration occurs in parallel with inflammation, even in the early phases of RRMS, with its intensity varying across different disease stages. Recent population studies show that the first nonspecific symptoms, such as fatigue, sleep disturbances, headaches, mood disorders, or gastrointestinal issues, appear in some patients five to ten years before a clinical diagnosis of MS is established (the so-called prodrome). These manifestations are probably the result of early immunological changes and subclinical inflammation in the CNS [24].

MS attacks represent the clinical manifestations of focal inflammatory damage to the central nervous system (CNS), most commonly in the brain’s white matter and spinal cord. A typical attack is defined as a new neurological deficit lasting for at least 24 h, without concurrent fever or other infections. The most common manifestations include unilateral optic neuritis (eye pain during movement, reduced visual acuity, and color vision disturbances), motor dysfunction (e.g., hemiparesis or paraparesis), sensory disturbances (hypoesthesia and paresthesia), and/or cerebellar dysfunction (vertigo, ataxia and intention tremor). Sphincter disturbances or sexual dysfunction due to the involvement of spinal tracts and the brainstem are also common [25].

MS is not a disease limited to episodes of inflammatory activity but also encompasses a broad spectrum of symptoms that significantly affect patients’ quality of life. Among the most important are psychiatric comorbidities, especially depression and anxiety disorders, whose prevalence in MS patients is much higher than in the general population. Depression affects approximately 30–50% of patients during the course of the disease, and is often underdiagnosed and insufficiently treated. Another symptom than that is difficult to treat is fatigue [26,27].

#### 2.1.1. Multiple Sclerosis and Reproduction

The topic of fertility and MS has been in the spotlight for a long time, and the number of studies devoted to this topic is increasing [28,29]. The negative impact of MS on reproduction is often considered, presented as a response to a chronic inflammatory process in the body, leading to downregulation of reproductive processes in an attempt to avoid a possible pregnancy, which would represent an additional burden for the body. Several factors leading to the development of MS and reduced fertility overlap (obesity, smoking, higher anxiety, antidepressant treatment, higher incidence of endometriosis, vitamin D deficiency, lower concentrations of anti-Müllerian hormone and motoric dysfunction) and some are MS specific (sexual dysfunction). However, the results of these studies have not proven that women with MS have reduced fertility [30]. According to a 2015 study from a French MS center, the time to pregnancy in MS patients is no different to that in the general population [31]. Before MS diagnosis, the average time was 8.57 months, and 7.53 months after MS diagnosis. The average number of children was 1.37. There was also no significant association between MS and the onset of menopause [32].

Some epidemiological studies have shown that women with MS have fewer children than those in the general population [33]. The cause is likely multifactorial. The link with endocrine changes, sexual dysfunctions including decreased libido and psychosocial factors including mood disorders, anxiety, and depression are discussed. Another reason for increased childlessness in MS patients may be concerns related to motherhood, transmission of the disease to offspring [34].

In women with MS, the ovaries may age faster due to immunosuppressive therapy [35]. In the past, a negative impact on the ovaries in MS patients associated with cytotoxic treatment (Mitoxanthrone or Cyclophosphamide) has been observed. However, this therapeutic method is currently not used, and the risk of ovarian damage is therefore significantly reduced [36,37].

#### 2.1.2. The Effect of MS on AMH Level

AMH is a hormone produced by Sertoli cells that prevents the development of the Müllerian duct in males during the prenatal period. In girls, it is undetectable prenatally, and its production occurs later with the onset of follicular growth. It is an important gonadotropin-independent marker of the ovarian reserve, and its level does not change significantly during the menstrual cycle. AMH is produced by granulosa cells of growing follicles (preantral to antral). AMH blocks the entry of primordial follicles into the growth stage, thus preventing the premature depletion of the follicular pool [38,39]. AMH also reduces the sensitivity of follicles to FSH, thereby inhibiting the development of additional antral follicles [40]. Maximum values in the blood are observed in the period after puberty, and from the age of 25, the levels gradually decrease to undetectable values during menopause. AMH levels are demonstrably reduced in women with systemic lupus erythematosus, an autoimmune disease [41]. The results of studies published to date on AMH levels in patients with MS are unclear. In particular, older clinical trials reported a decrease in AMH values in patients with MS, but this has not been proven in more recent studies [42,43,44]. It has been described that AMH levels in women with MS are the same as in women without MS, and the age of menopause does not differ [45,46]. In a large meta-analysis, it was reported that AMH levels in women with MS were not significantly lower than those in healthy women. However, lower estradiol levels, lower antral follicle count (AFC) values, and higher LH levels were found [47]. This result is interesting; however, the meta-analysis did not consider the stage of the disease or its therapy, which may be misleading, especially since AMH levels can be affected by the course of the disease and the MS therapy used. Previous studies have described a decrease in AMH levels in women with MS [48], probably due to immunosuppressive treatment-induced ovarian damage. A large study from 2018 focused on patients with MS in whom AMH values were lower than in other women, and this condition was associated with a higher rate of progression, radiological findings, and clinical disability [44]. It appears that AMH levels can be significantly affected by the course of MS. Sepúlveda et al. (2015) [49] observed a different ovarian reserve in women with MS in relation to the number of relapses per year and a negative correlation between disease activity and AMH levels and AFC. Another study examining ovarian reserve, including AMH levels, demonstrated that women with higher MS disease activity had significantly lower AMH levels, reduced total antral follicle count, and reduced ovarian volume compared to women with lower disease activity [49].

#### 2.1.3. The Influence of Sex Hormones on the Development and Course of MS

Hormones play an important role in the course of the disease, which is demonstrated not only by the different courses of MS in different periods of life (postpubertal age, pregnancy, breastfeeding, and menopause), but also by the different courses of the disease in men and women. Sex hormone levels play an important role in the etiopathogenesis and course of MS because of the significant influence of the immune response, depending on their concentration [50].

#### 2.1.4. Menstrual Cycle

Fluctuations in the severity of neurological symptoms can also occur during the menstrual cycle. A temporary worsening of MS symptoms often occurs before the menstruation [50]. The most common difficulties include increased fatigue, weakness, imbalance, and depression. This may be due to a sudden decrease in estradiol and progesterone levels before the onset of menstrual flow.

#### 2.1.5. Pregnancy

The understanding of all immune processes during pregnancy in patients with MS is certainly not complete, but we are still learning more information not only about hormonal changes but also about the immune mechanisms. Fetal antigens affect the maternal immune system. In the first trimester, pro-inflammatory activity is necessary for the implantation of the blastocyst into the uterus. Physiologically, in the early phase after fertilization, the maternal organism’s immunotolerance towards the fetus is rapidly established, the activity of Th1 lymphocytes decreases, and the Th2 lymphocyte response predominates, leading to the production of cytokines (IL-4, IL-5, IL-6, IL-10), which suppress the excessive immune response and facilitate the implantation of the embryo in the uterus [51]. All of these contribute to the alleviation of MS symptoms, with a maximum in the third trimester when the concentrations of sex hormones are highest. In contrast, their sudden decrease immediately after birth reverses the immune response in favor of a pro-inflammatory response, as a logical protection for the weakened and infected mother to recover. This explains the higher percentage of MS relapses during the early postpartum period. During exclusive breastfeeding, prolactin levels are high, and GnRH and LH release are suppressed, leading to amenorrhea and a protective role against increased inflammatory activity [52].

#### 2.1.6. MS Therapy and Possible Risks of ART

The treatment of MS is based on two main principles: acute attack therapy and long-term immunomodulatory therapy. Acute attacks are usually treated with intravenous corticosteroids (e.g., methylprednisolone 1000 mg daily for 3–5 days), which suppress inflammatory activity and accelerate the resolution of symptoms. In severe or corticosteroid-resistant cases, plasmapheresis or intravenous high-dose immunoglobulin therapy is considered. Long-term treatment aims to reduce the number of relapses, slow the progression of disability, and reduce inflammatory activity on magnetic resonance imaging (MRI). There is a full spectrum of available disease-modifying drugs, including injectables (interferons, glatiramer acetate), oral drugs (e.g., dimethyl fumarate, teriflunomide, fingolimod), and currently preferred highly effective therapy (e.g., natalizumab, anti-CD20 antibodies, cladribine). The current therapeutic approach is guided by the No Evidence of Disease Activity (NEDA) concept, which includes the absence of clinical attacks, a stable neurological score and no new activity on MRI. The aim is to achieve maximum disease control at an early stage, ideally by using early high-efficacy therapy in patients at risk. Regular clinical and radiological monitoring is key for the early detection of treatment failure and adjustment of the strategy [53].

#### 2.1.7. Pregnancy and Treatment of MS

Today we know that patients with MS can become pregnant and at the same time be effectively treated. Appropriate long-term therapy for MS should be selected individually, including according to the time of the planned pregnancy (whether it is a current topic or the pregnancy is planned months to years in the future). Care should be taken with drugs that cannot be used during pregnancy or breastfeeding. Currently, there are drugs available that can be administered throughout pregnancy and breastfeeding, or pulse therapy that ensures the duration of pregnancy and breastfeeding without the need for medication. It is more appropriate to plan pregnancy when the patient is clinically and radiologically stable [54].

### 2.2. Assisted Reproduction Methods in Relation to MS

Major changes in sex hormone levels occur during infertility treatment using ART techniques. After IVF treatment, hormone levels decrease. This period is associated with an increased risk of exacerbation of autoimmune diseases, including MS [55]. Simultaneously, psychological stress during the IVF cycle and subsequent anxiety in the case of failure can contribute to immune dysregulation, leading to the outbreak of an autoimmune disorder [56].

#### 2.2.1. The Impact of ART on the Risk of Developing MS

Cases of MS development during or after IVF therapy have been reported [57]. For example, in 2015, a case was presented of a 36 year old woman who developed right-sided hemiparesis after her fifth IVF cycle (with a corresponding demyelination correlate on brain MRI) and a high risk of developing definitive MS [58]. These are serious cases, but in recent years, the occurrence of similar cases has become rare. Although the reported cases were severe, they were isolated case reports and were not representative of the entire population. Large multicenter meta-analyses are more significant than individual cases and provide a more accurate picture of the actual situation. In 2022, a large Danish study focusing on the incidence of MS in women who underwent infertility treatment using ART was presented. This study included 585,716 women, of whom 11% (63,791) had at least one initiated IVF cycle. The data were reported from the Danish registry between 1996–2018, and the final results were not analyzed with regard to the cause of infertility. Women who underwent ART were older than those without IVF (31.8 vs. 27.5 years). No association was found between the incidence of MS and exposure to ART compared with pregnancies without ART. When comparing the ART cycle subgroups, no difference was found between successful (pregnancy) and unsuccessful cycles in terms of the increased risk of MS within two years of starting the cycle. Based on these studies, women treated with ART are not at a higher risk of developing MS than those who did not undergo ART [59].

#### 2.2.2. The Impact of ART on the Course of MS

##### Older Studies: Increased Risk of Relapses

Earlier studies published between 1998 and 2012, when highly effective therapies (HET) were less available, observed an increase in MS relapses following ART. For example, Hellwig et al. (2008) observed a 33% increase in the annual relapse rate in the first three months after an unsuccessful IVF cycle in 32 patients [60]. Similar results were reported by Michel et al. (2012) who identified interruption of immunomodulatory therapy, hormonal stimulation, and stress associated with the procedure as risk factors [61].

##### New Insights: Stabilization Thanks to Modern Therapy

More recent studies published after 2015 have shown a different trend. With the introduction of HET (e.g., natalizumab, fingolimod, anti-CD20 antibodies) and the shift to individually planned pregnancies in stabilized patients, the incidence of relapse after ART has decreased. For example, a multicenter retrospective study [62] including 124 ART cycles in patients with MS or neuromyelitis optica spectrum disorder found stable or low disease activity in most patients, especially when HET was maintained or briefly interrupted.

A recent review by Mainguy summarized both earlier small single-center studies (2006–2012) and newer studies with larger sample sizes. The original warnings about the increased risk of relapse after IVF have not been confirmed in current studies with larger patient cohorts [29]. Recent prospective data from German registries (AAN 2024) show that women with MS who continue specific immunomodulatory therapy during ART have a significantly lower risk of relapse (5.9% vs. 16.7%, *p* = 0.009) [63]. In another study examining the relapse rate after ART procedures, it was found that the relapse rate was significantly higher in the three months following ART than in the previous 12 months. It is possible that the increased MS activity after ART may be triggered by a sudden drop in estrogen levels, which occurs after ART failure, similar to what occurs after miscarriage or normal childbirth [62]. In 2023, the results of a multicenter study focusing solely on patients with MS who underwent infertility treatment were published. This study included four academic MS centers, where a total of 65 patients who underwent 124 treatment cycles were analyzed. Patients were monitored for 12 months prior to and for three months after therapy. Of the 80 controlled ovarian stimulation cycles, only five relapses occurred in four patients. The overall relapse rate after infertility treatment did not increase compared to that before therapy. The monitored therapies included ovarian stimulation with embryo transfer, stimulation without embryo transfer, embryo transfer only, and induction of ovulation. No relationship was found between MS relapse and treatment type or therapeutic protocol [64].

A graphical representation of the recommended treatment approach for infertility in patients with multiple sclerosis is shown in Figure 2.

MS and Methods of Ovarian Stimulation: During ovarian stimulation, multiple antral follicles grow simultaneously, leading to increased estrogen production, with serum estrogen levels rising dramatically over a short period of about 14 days. In the past, studies have reported increased disease activity in MS patients undergoing ART protocols based on GnRH hormone agonists [9,65]. The underlying mechanism may be related to the concentration of estrogen. High estrogen concentrations transform Th1 cells into Th2 cells, decrease the number of Th17 cells, and activate Treg cells [66]. An imbalance among certain T-cell types contributes to MS relapse [67].

Currently, there are options to modify these conditions using selective modulators. Aromatase inhibitors, such as Letrozole, cause the downregulation of estrogens by inhibiting cytochrome P450, thereby increasing FSH production. This approach is, used, for example, in the treatment of women with estrogen-sensitive cancers [68]. These forms of ovarian stimulation (without GnRH agonists) are currently safer and recommended for MS patients. The safety of these stimulation protocols has been supported by recent studies [59,64,69].

However, there have also been cases in the past where definitive MS developed after IVF. In 2016, two cases were described (aged 36 and 29) where MS clinically manifested following the initiation of IVF. The 36 year old underwent an antagonist protocol (ganirelix, menotropin, follitropin alfa, and choriogonadotropin alfa), while the 29 year old received an agonist protocol (follitropin alfa, lutropin alfa, and choriogonadotropin) [57]. Both women were healthy, with no significant family history, and had never exhibited MS symptoms before. Such cases are currently very rare, and objective assessment of ART’s impact on MS requires long-term monitoring of populations involving as many patients as possible a goal aided today by nationwide or global patient registries.

In France, a multicenter study conducted in 2012 included 32 women with MS who underwent 70 IVF treatments (48 agonist protocols, 19 antagonist protocols). The study found a significant increase in the annualized relapse rate (ARR) during the three-month period after IVF. The increased risk of relapse was observed not only in agonist protocols (explained by direct immunological effects, longer stimulation, and greater fluctuations in gonadotropins and steroids), but also in cases where IVF was unsuccessful. No increased relapse risk was found with antagonist protocols [61].

Torkildsen et al. reported a severe relapse in a patient with relapsing-remitting MS two months after stopping fingolimod and three days after starting stimulation with Follistim (375 IU) and Menopur (100 IU), thus without using GnRH agonists [70]. The authors themselves emphasize that the increased risk of disease activity is not solely due to stimulation; the influence of other factors, including discontinued medication, cannot be excluded. Most typically, a rebound phenomenon or relapse develops eight to nine weeks after fingolimod cessation, as demonstrated by data from the Finnish registry [71]. The patient’s stimulation in this case report took place approximately during the period of highest risk for disease flare-up. Table 1 presents a summary of small retrospective case reports and cohort studies focusing on stimulation methods in MS patients conducted in recent years.

In other therapies using GnRH agonists and MSA, relapse after using GnRH agonists has also been observed in the treatment of other diagnoses requiring these medications, such as endometriosis or uterine fibroids. In 2020, a case was reported involving a 39 year old woman diagnosed with uterine fibroids. After starting treatment with a GnRH agonist (leuprorelin acetate; 30 μg/kg subcutaneous injection every 4 weeks), she experienced various neurological deficits. After four cycles of this treatment, she developed severe neurological symptoms and was diagnosed with MS [72]. However, this is an isolated case of newly diagnosed MS after GnRH agonist therapy. This patient may have had subclinical MS prior to GnRH agonist treatment.

#### 2.2.3. Oocyte Cryopreservation (OC)

In recent times, the practice of vitrifying oocytes has extended beyond oncological patients to include those with endometriosis or autoimmune disorders [73]. For women requiring assisted reproduction who also have multiple sclerosis, preserving fertility through oocyte cryopreservation can be a viable option. Similar to IVF, the OC process involves administering drugs to stimulate the ovaries. Mature oocytes are collected through transvaginal aspiration of antral follicles in the operating room. These oocytes are then vitrified and stored in liquid nitrogen, allowing for long-term preservation. However, the benefits of oocyte freezing are contingent upon certain conditions. A critical factor is the woman’s age, as it is not practical to store oocytes from women over 40 due to the rapid decline in quality with age. OC is advantageous only for women with a sufficient number of oocytes that can be retrieved with a reasonable number of stimulations. Therefore, women should have a moderately good ovarian reserve, assessed by AMH levels and the count of antral follicles. Women meeting these criteria can proceed with oocyte freezing. In cases where they need to delay motherhood for about a year, perhaps due to uncontrolled MS, short-term DMT use, or a long DMT washout period, they will have oocytes stored with the same quality as before vitrification. Given the rapid decline in oocyte quality in women over 30, this is a compelling reason to consider OC [74].

#### 2.2.4. Embryo Cryopreservation (EC)

For EC, the oocytes are fertilized, and the embryos are vitrified on the fifth day of in vitro development. The vitrification process is more effective in embryos than in oocytes. In some cases, this can be a good step for preserving fertility in patients with MS, but sometimes, the vitrification of human embryos encounters ethical limits that can complicate this method of fertility preservation.

### 2.3. Recommendations for the Management of Assisted Reproduction in Women with Multiple Sclerosis

Pre-conception Counseling & Disease Stabilization
Women with MS should undergo **individualized pre-conception counseling** involving a multidisciplinary team (neurologist, gynecologist, embryologist, radiologist).ART should be planned **only when MS is clinically and radiologically stable**, ideally under a modern DMT regimen.**Do not discontinue DMT prematurely**; abrupt cessation—especially of fingolimod or natalizumab—significantly increases relapse risk.
Ovarian Reserve Assessment & Fertility Preservation
Evaluate ovarian reserve using AMH levels and antral follicle count before initiating ART [68].Discuss fertility preservation options (oocyte or embryo cryopreservation) early, particularly in:
∘young women not yet ready for pregnancy∘those requiring DMTs with long washout periods∘patients with expected disease progression

Selection of Ovarian Stimulation Protocol
Prefer GnRH antagonist protocols or letrozole-based stimulation, which are associated with lower relapse risk than older GnRH agonist protocols.Avoid high-estrogen stimulation approaches in unstable MS unless strongly indicated.Minimize treatment-induced hormonal fluctuations, which may trigger MS activity.Use follitropin alpha/beta or delta for ovarian stimulation.Use GnRH agonist for final oocyte maturation in women seeking fertility preservation for medical reasons [75].
ART Procedure and Monitoring
Maintain DMT when clinically justified; **continued or minimally interrupted therapy** reduces relapse risk during ART.Monitor Neurological Symptoms Closely During:
∘ovarian stimulation∘postoperative period∘and the 3 months following unsuccessful cycles, when relapse risk is highest

Management After Embryo Transfer and During Pregnancy
Once pregnant, continue care in an interdisciplinary setting.Select pregnancy-compatible DMTs for women at high relapse risk.Recognize the **postpartum period** as a vulnerable phase; plan proactive monitoring and early postpartum management.
Shared Decision-Making
Tailor ART and MS therapy decisions to:
∘disease activity∘age and ovarian reserve∘reproductive goals∘and prior ART outcomes
Engage the patient in shared decision-making to balance fertility desires with disease control and treatment safety (summarized in Table 2).


### 2.4. Future Perspectives

In the future, to improve the quality of infertility treatment in MS patients, more data is needed on modern MS drugs and their use before planned pregnancy and during breastfeeding. To make current practices more effective, it is necessary to have more data on healthy lifestyle factors that can positively influence planned pregnancy and MS (vitamin D, microbiome, physical activity). In high-risk cases when it is helpful to avoid ovarian stimulation, it can be beneficial to use specific cryopreservation methods, such as freezing immature oocytes or ovarian tissue for fertility preservation, as this is important for supporting the next stage of advanced cryopreservation technics.

### 2.5. Limitations

A limiting factor in assessing the effect of IVF on patients with MS is the limited number of studies focusing on the analysis of ART in people with MS. The main limitation of our recommendations is that they are based primarily on the personal experiences of specialists with assisted reproductive therapy in women without multiple sclerosis. However, a recent large population study found that the chance of having a healthy child after ART in women with MS was similar to that in women without MS [76]. However, further research is needed to analyze the impact of ART techniques on some important aspects of MS (e.g., relapses, disability scores, potential effects, and interactions of corticosteroids and other DMTs during ART). Our recommendation is a consensus of Czech neurologists and gynecologists, and we believe that it should be valid in other European countries as well, although the specific environment in regions other than the Czech Republic may limit its practical use.

## 3. Conclusions

Contemporary approaches to multiple sclerosis therapy do not preclude women from pursuing pregnancy or utilizing in vitro fertilization methods if necessary. Recent studies indicate that women undergoing assisted reproduction are not at an elevated risk of developing MS compared to those who do not undergo IVF. Furthermore, the use of modern, highly effective drugs may also contribute to the overall stabilization of the patient and lower inflammatory activity in MS, even during infertility treatment.

## Figures and Tables

**Figure 1 healthcare-13-03155-f001:**
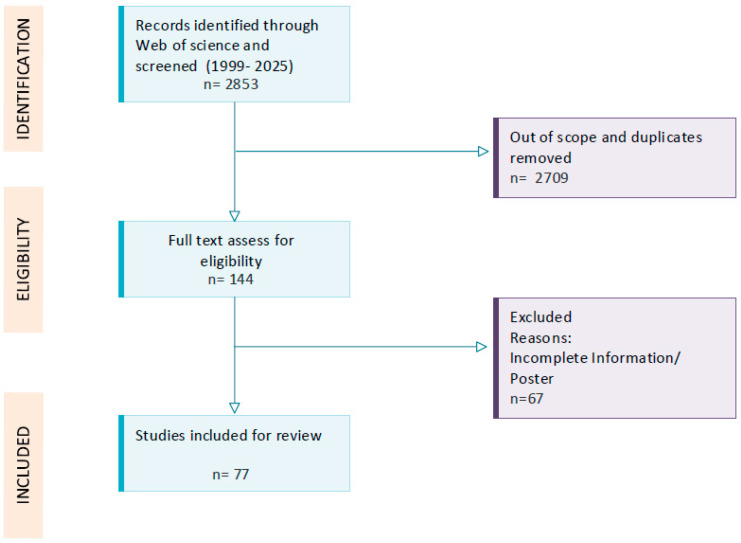
Review flow diagram of the study selection and record screening process. Publications from 1999 to 2025 were selected according to key words and their eligibility for this review. Finally 77 studies were included.

**Figure 2 healthcare-13-03155-f002:**
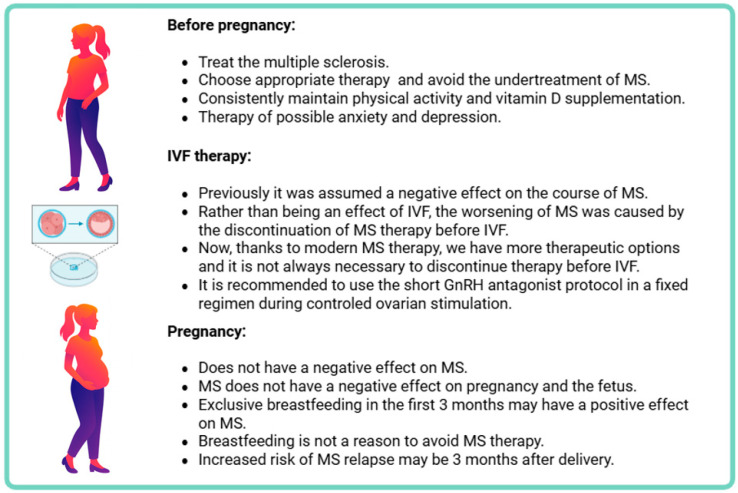
Management of pregnancy and IVF therapy in patients with multiple sclerosis. During infertility therapy in MS patients, it is important to distinguish between the condition before IVF treatment, IVF itself, and the treatment of pregnant patients after successful IVF treatment.

**Table 1 healthcare-13-03155-t001:** Studies have focused on ART in multiple sclerosis patients. This table highlights the type of stimulation and relapse rate in patients after ovarian stimulation.

Study	Year	Study Design	Sample Size/ART Cycles	Reaps Rate (RR) GnRH Agonist	After ART GnRH Antagonist	Notes	Country
Hellwig et al. [60]	2008	R, P	23/78	Increased	Increased	RR increased independently on the intervals between	Germany
Michel et al. [61]	2012	R	32/70	Increased	Not-increased	RR increased after ART failure	France
Correale et al. [65]	2012	P	16/26	Increased	Non included	ninefold increased in the risk of MRI activity	Argentina
Bove et al. [62]	2020	C, MA	12/22	Not-increased	Not-increased	Small study, higher RR when topped treatment > 3 months before VF than patients which continued in ART	USA
Mainguy et al. [29]	2025	R	115/199	Not-increased	Not-increased	Lower RR after ART taking DMTS	France
Graham et al. [64]	2023	R	65/124	Not-increased	Not-increased	RR increased after 2 or more stimulations	USA
Kopp et al. [59]	2023	C, MA	585,716/63,791	Not-increased	Not-increased	non significant trend toward increased risk of MS with higher number of ART cycles	Denmark

C—cohort study, R—retrospective study. P—prospective study, MA—meta analysis.

**Table 2 healthcare-13-03155-t002:** The table contains essential points in the management of infertility therapy in patients with MS.

Category	Key Recommendations
Pre-conception Counseling & Disease Stabilization	Individualized counseling; plan ART only when MS is stable; avoid premature DMT discontinuation.
Ovarian Reserve Assessment & Fertility Preservation	Assess AMH and AFC; discuss oocyte/embryo freezing especially in young women or those on long washout DMTs.
Selection of Stimulation Protocol	Prefer GnRH antagonist or letrozole-based protocols; avoid high-estrogen stimulation in unstable MS.
ART Procedure and Monitoring	Maintain DMT when justified; monitor closely during stimulation, post-op, and 3 months after unsuccessful cycles.
Management After Embryo Transfer & Pregnancy	Continue interdisciplinary care; select pregnancy-compatible DMTs; monitor proactively postpartum.
Shared Decision-Making	Tailor ART to disease activity, age, ovarian reserve, and patient goals.

## Data Availability

No new data were created or analyzed in this study.

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
