# Peer review of "Assisted Reproduction Therapy in Patients with Multiple Sclerosis: Narrative Review and Practical Recommendations"

_healthcare, 2025, doi:10.3390/healthcare13233155_

Round 1

Reviewer 1 Report (Previous Reviewer 1)

Comments and Suggestions for Authors

The revision has been done properly. it is publishable

Author Response

Comments:The revision has been done properly. it is publishable.

Response: Thank you for your time for reviewing our manuscript and your positive reaction. We really appreciate it.

Reviewer 2 Report (Previous Reviewer 2)

Comments and Suggestions for Authors

This his study have has  is to present contemporary findings regarding the relationship between the application of assisted reproduction methods and their impact  on the incidence of multiple sclerosis. It is a Design study elaborated as narrative review article with a focus does on health risk in a IVF treatment in this group of patients. Therefore, it is necessary to always individualize therapy with regard to future 28 pregnancy. Interdisciplinary cooperation on the timing of IVF therapy and minimizing 29 the risk of MS exacerbation is also important.

The scientific work helps the reader understand the specific characteristics of this group of patients and their doubts and health requirements in the context of assisted reproductive technology (ART). The work is comprehensive in its narrative, with explanatory figures and tables reviewing the limited literature available. The English is well written. I would say that the work is interesting and offers a useful insight for practitioners seeking advice prior to ART treatments. It is publishable.

Author Response

Comments: This his study have has  is to present contemporary findings regarding the relationship between the application of assisted reproduction methods and their impact  on the incidence of multiple sclerosis. It is a Design study elaborated as narrative review article with a focus does on health risk in a IVF treatment in this group of patients. Therefore, it is necessary to always individualize therapy with regard to future 28 pregnancy. Interdisciplinary cooperation on the timing of IVF therapy and minimizing 29 the risk of MS exacerbation is also important.

The scientific work helps the reader understand the specific characteristics of this group of patients and their doubts and health requirements in the context of assisted reproductive technology (ART). The work is comprehensive in its narrative, with explanatory figures and tables reviewing the limited literature available. The English is well written. I would say that the work is interesting and offers a useful insight for practitioners seeking advice prior to ART treatments. It is publishable.

Response: Thank you for your positive evaluation. We really appreciate it.

Reviewer 3 Report (Previous Reviewer 3)

Comments and Suggestions for Authors

The manuscript addresses an important and underexplored topic at the intersection of neurology and reproductive medicine: the implications of assisted reproductive technology (ART) in women with multiple sclerosis (MS). The review is timely and clinically relevant, considering the increasing prevalence of both infertility treatments and MS in reproductive-age women. The interdisciplinary perspective adopted by the authors is commendable, as it integrates gynecological, neurological, and embryological viewpoints.

However, while the manuscript provides a broad and detailed overview of existing literature, certain sections would benefit from improved clarity, conciseness, and analytical depth. Some structural and stylistic refinements would also enhance readability and scientific rigor.

The introduction could better emphasize the clinical gap that motivates the review (i.e., absence of specific ART guidelines for MS patients).

The authors should define abbreviations upon first use consistently (e.g., DMT, ART, MS).

The paper addresses an important multidisciplinary topic and offers valuable practical insights. However, the authors should revise the manuscript to:

  • Clarify methodological rigor,

  • Reduce redundancy,

  • Refine the language for academic precision.

Once these revisions are addressed, the manuscript will likely meet publication standards.

Authors should correct minor grammatical errors: inconsistent capitalization (e.g., “multiple sclerosis” vs. “Multiple Sclerosis”), (the phrase "For this is importent to support next progress of aadvanced cryopreservation technics")and occasional awkward phrasing detract from clarity.

Author Response

Comments1: The manuscript addresses an important and underexplored topic at the intersection of neurology and reproductive medicine: the implications of assisted reproductive technology (ART) in women with multiple sclerosis (MS). The review is timely and clinically relevant, considering the increasing prevalence of both infertility treatments and MS in reproductive-age women. The interdisciplinary perspective adopted by the authors is commendable, as it integrates gynecological, neurological, and embryological viewpoints.

However, while the manuscript provides a broad and detailed overview of existing literature, certain sections would benefit from improved clarity, conciseness, and analytical depth. Some structural and stylistic refinements would also enhance readability and scientific rigor.

Response 1: The manuscript has been revised and improved.

Comments2: The introduction could better emphasize the clinical gap that motivates the review (i.e., absence of specific ART guidelines for MS patients).

Response 2: The introduction was suplemented and new sentences were added.

Comments3: The authors should define abbreviations upon first use consistently (e.g., DMT, ART, MS).

Response 3: Abbreviatons was revised and corrected.

Comments4: The paper addresses an important multidisciplinary topic and offers valuable practical insights. However, the authors should revise the manuscript to:

Clarify methodological rigor,

Response 3: Methods were changed and suplemented

Reduce redundancy,

All redundancy was reduced

Refine the language for academic precision.

The English throughout the manuscript has been checked and corrected.

Once these revisions are addressed, the manuscript will likely meet publication standards.

Comments5: Authors should correct minor grammatical errors: inconsistent capitalization (e.g., “multiple sclerosis” vs. “Multiple Sclerosis”), (the phrase "For this is importent to support next progress of aadvanced cryopreservation technics")and occasional awkward phrasing detract from clarity.

Response 5: Minor errors were corrected.

Thank you for reviewing our manuscript, which has been significantly revised and supplemented and we believe it is now more suitable for publication.

Reviewer 4 Report (Previous Reviewer 4)

Comments and Suggestions for Authors

Major concerns:

  1. The whole first paragraph on introduction contains several distinct ideas, and not a single citation. Please revise.
  2. Please reiterate in the aim of study that this is a Narrative Review.
  3. In methods mention between which years have you extracted the articles. Did you go back 10, 20 years or more?
  4. Figure 1, and ll figures for the matter, mush be named and shortly introduced/explained in text beforehand. This is a PRISMA -like flowchart. Because this is a Narrative Review, you don't require this in theory. Nevertheless, I appreciate it. Please mention this info and explain why you choose to include it. Furthermore, all tables and figures demand a short explanatory footnote.
  5. Did you completely skip ethical considerations? I do understand this is a retrospective review, but, for the very least, please mention that it was carried out in accordance to the declaration of Helsinki (10.1001/jama.2013.281053 ) and it respects European GDPR (10.1007/978-3-319-99713-1_5). Was this carried out in a research facility under the jurisdiction of an academic body? Please revise.
  6. In Etiopathogenesis (?):  Second paragraph has no citation. Please add. Third paragraph has only one citation, please revise.
  7. Peripheral Activation of the Immune System subchapter has no citations. Please add. ??? 
  8. This trend continues over the course of the subchapter "Inflammation and Damage in the Central Nervous system;" and further in the whole article. Either there is only one citation at the end of the paragraph - highly skeptical of it's comprehensiveness - or not at all. This is not compatible with a published material and must be thoroughly addressed.
  9. "New insights: stabilization thanks to modern therapy" subchapter is doubled. Please revise.
  10. Figure 2 - in text introduction and footnote required.
  11. Table 1 - same.
  12. "Recommended Procedure for Ovarian Stimulation in Women with MS" reiterates information already presented and does not summarize efficiently any procedure or protocol. This begs the question - What is the new insight that this manuscript offers? This is a major concern and should be addressed proactively. I suggest giving clinical practical pointers, highlight suggested protocols, offer diagrams or figures for highlighting them and maybe this would be sufficient to push this into a publishable material, in regards to novelty and bringing something new to the table.
  13. I commend you for having limitations and future perspectives.

Looking forward to your response. Good luck.

Author Response

Comments1: Major concerns:

The whole first paragraph on introduction contains several distinct ideas, and not a single citation. Please revise.

Response 1: Paragraph was revised and citation was added.

Comments2: Please reiterate in the aim of study that this is a Narrative Review.

Response 2: It was corrected.

Comments3: In methods mention between which years have you extracted the articles. Did you go back 10, 20 years or more?

Response 3: Thank you for this point now is it correct.

Comments4: Figure 1, and ll figures for the matter, mush be named and shortly introduced/explained in text beforehand. This is a PRISMA -like flowchart. Because this is a Narrative Review, you don't require this in theory. Nevertheless, I appreciate it. Please mention this info and explain why you choose to include it. Furthermore, all tables and figures demand a short explanatory footnote.

Response 4: Figures and tables are now introduced in text.

Comments5: Did you completely skip ethical considerations? I do understand this is a retrospective review, but, for the very least, please mention that it was carried out in accordance to the declaration of Helsinki (10.1001/jama.2013.281053 ) and it respects European GDPR (10.1007/978-3-319-99713-1_5). Was this carried out in a research facility under the jurisdiction of an academic body? Please revise.

Response 5: Ethical aspect was included in this manuscript.

Comments6: In Etiopathogenesis (?):  Second paragraph has no citation. Please add. Third paragraph has only one citation, please revise.

Response 6: Chapter etiopathogenesis was update and new citations was added.

Comments7: Peripheral Activation of the Immune System subchapter has no citations. Please add. ???

Response 7: This chapter was changed and new citation was added.

Comments8: This trend continues over the course of the subchapter "Inflammation and Damage in the Central Nervous system;" and further in the whole article. Either there is only one citation at the end of the paragraph - highly skeptical of it's comprehensiveness - or not at all. This is not compatible with a published material and must be thoroughly addressed.

Response 8: New citations was added into this chapter.

Comments9: "New insights: stabilization thanks to modern therapy" subchapter is doubled. Please revise.

Response 9: Doublet part was deleted.

Comments10: Figure 2 - in text introduction and footnote required.

Table 1 - same.

Response 10: Introduction and footnote for Figure 2 and Table 1 was added.

Comments11: "Recommended Procedure for Ovarian Stimulation in Women with MS" reiterates information already presented and does not summarize efficiently any procedure or protocol. This begs the question - What is the new insight that this manuscript offers? This is a major concern and should be addressed proactively. I suggest giving clinical practical pointers, highlight suggested protocols, offer diagrams or figures for highlighting them and maybe this would be sufficient to push this into a publishable material, in regards to novelty and bringing something new to the table.

Response 11: Recommendaton was completely changed. Now is it in points and key points are also visible in table.

I commend you for having limitations and future perspectives.

Looking forward to your response. Good luck.

Thank you for reviewing our manuscript, which has been revised and supplemented following your comments and we believe that it is now more suitable for publication.

Round 2

Reviewer 3 Report (Previous Reviewer 3)

Comments and Suggestions for Authors

The recommended changes have been implemented and the manuscript has improved significantly.

Comments on the Quality of English Language

English quality is fine. 

This manuscript is a resubmission of an earlier submission. The following is a list of the peer review reports and author responses from that submission.

Round 1

Reviewer 1 Report

Comments and Suggestions for Authors

The manuscript addresses an important and clinically relevant intersection between assisted reproductive technology (ART) and multiple sclerosis (MS). However, despite the significance of the topic, the paper in its current form suffers from several fundamental weaknesses in terms of structure, scientific depth, and originality. These issues considerably limit its contribution to the literature and, in my opinion, do not support publication in its present form.

1) The manuscript largely repeats well-established knowledge from prior reviews and consensus papers, without offering new perspectives, data, or critical analysis.

2) The search strategy described (PubMed, WoS, Scopus) is insufficiently detailed. There is no mention of search terms, inclusion/exclusion criteria, study selection flow, or quality assessment.

3) Tables, figures, or schematic summaries would have been helpful but are absent.

Reviewer 2 Report

Comments and Suggestions for Authors

The objective of this review is to present  findings regarding the relationship between the application of ART methods and their impact on multiple sclerosis. The work provides an important description of the studies published in the literature with an impact factor, with an examination ranging from the aetiopathogenesis of the disease to how it should be investigated in centres where ART is performed, up to the subsequent pregnancy. I find the work satisfactory, well written, very useful for readers who come into contact with this subset of patients, and prepares them to reassure them by providing evidence of the safety of ART treatments: the initiation of highly effective MS therapy does not result in increased inflammatory activity or exacerbation of the disease course in these patients during infertility treatment The English is well written and I would say that the article is publishable as presented, thanks also to a complete and satisfactory bibliography.

Reviewer 3 Report

Comments and Suggestions for Authors

This manuscript presents a comprehensive narrative review on the safety and implications of assisted reproductive technology (ART) in women with multiple sclerosis (MS). The topic is timely and clinically relevant, given the increasing prevalence of both MS and ART use among women of reproductive age. The paper provides a broad overview of hormonal, immunological, and clinical aspects and integrates findings from recent studies.
Overall, the article is informative, up-to-date, and multidisciplinary, but it requires improvements in several specific areas:

-The extensive background on MS pathogenesis and immunology, while accurate, occupies a large portion of the manuscript and detracts from the review’s main focus. Please, consider condensing this section and emphasizing findings that directly relate to ART and reproductive endocrinology.

- The conclusion that ART is entirely safe for women with MS may be somewhat overstated. Although recent large-scale studies show no increased risk of MS onset or relapse, smaller cohorts have reported relapse clusters post-ART, particularly with hormonal fluctuations. A more nuanced conclusion acknowledging these residual uncertainties would strengthen credibility.

Minor Comments:

-Language and style: the text is generally clear but could benefit from editing for conciseness and avoidance of redundancy.

-Terminology: ensure consistent use of abbreviations (e.g., ART, DMT, MS).

Implementing these revisions will enhance scientific rigor and substantially improve the article.

Reviewer 4 Report

Comments and Suggestions for Authors

Hello.

Major concerns:

  1. The title should contain the type of study (e.g. narrative review / systematic review).
  2. That being said, which one is it? I presume narrative? 
  3. Where are the methods / study selection process chapter?
  4. I do understand somehow that the whole structure can mimic the discussion chapter, but you still require a Limitations and Future Directions subchapter.
  5. The introduction is lackluster, it does not cover most points it should. What is the aim of the study?
  6. What is the original part that is being added by this review?
  7. All those being said, the manuscript does not follow any kind of structure and lacks mandatory methodological focus points. 

I appreciate your work. This is not publishable like this. It contains interesting info, but it is not a research paper at the moment - more of a course. I recommend reading into the structure of a review, as a starting point for your journey. After that, I advise a reupload of the improved manuscript, to be able to put your work to good use. I wish you good luck.